



# Highly Stable Magic Angle Spinning Spherical Rotors Lacking Turbine Grooves

Thomas M. Osborn Popp[1], Alexander Däpp[1], Chukun Gao[1], Pin-Hui Chen[1], Lauren E. Price[1], Nicholas H. Alaniva[1], and Alexander B. Barnes[1]

[1]Laboratory for Physical Chemistry, ETH Zürich, Vladimir-Prelog-Weg 2, 8093 Zürich, Switzerland

**Correspondence:** Alexander Barnes (alexander.barnes@phys.chem.ethz.ch)

**Abstract.** The use of spherical rotors for magic angle spinning offers a number of advantages including improved sample exchange, efficient microwave coupling for dynamic nuclear polarization nuclear magnetic resonance (NMR) experiments and, most significantly, high frequency and stable spinning with minimal risk of rotor crash. Here we demonstrate the simple retrofitting of a commercial NMR probe with MAS spheres for solid-state NMR. We analyze a series of turbine groove
geometries to investigate the importance of the rotor surface on spinning performance. Of note, rotors lacking any surface modification spin rapidly and stably even without feedback control. The high stability of a spherical rotor about the magic angle is shown to be dependent on its inertia tensor rather than the presence of turbine grooves.

## 1 Introduction

Magic angle spinning (MAS) nuclear magnetic resonance (NMR) is usually used for high resolution analysis of the local
chemical environments of nuclear spins within biomolecular and inorganic solids (Schaefer and Stejskal (1976); McDermott (2009); Doty and Ellis (1981); Knight et al. (2012); Retel et al. (2017); Theint et al. (2017); Petkova et al. (2005); Kong et al. (2013); Wang et al. (2013); Cegelski et al. (2002); Bougault et al. (2019); Clauss et al. (1993); Trebosc et al. (2005); Lesage et al. (2008)). The sample is spun rapidly about an axis inclined at the magic angle, which is 54.74° with respect to the external magnetic field $B_0$. This averages terms in the NMR Hamiltonian whose orientational dependence is described by the second-
order Legendre polynomial $3\cos^2\theta - 1$ (Andrew et al. (1959); Lowe (1959); Andrew (1981)). For spins-1/2, MAS can yield spectra with highly resolved isotropic chemical shifts.

MAS has traditionally been performed by spinning a cylindrical rotor within a stator installed at the magic angle, thereby requiring a gas bearing to stabilize the rotor and drive gas to apply torque (Andrew (1981); Doty and Ellis (1981); Wilhelm et al. (2015)). However, we recently showed that it is possible to spin samples via a different paradigm, namely using spherical
rotors spun using a single gas stream for both the bearing and drive (Chen et al. (2018); Gao et al. (2019)). This approach allows highly stable rotor spinning about a single axis inclined at the magic angle, with record rates as high as 4.6 kHz ($N_2$, 4.1 Bar) and 10.6 kHz (He, 11 Bar) for 9.5 mm diameter rotors and 11.4 kHz ($N_2$, 3.1 Bar) and 28 kHz (He, 7.6 Bar) for 4 mm diameter rotors. Decreasing the rotor diameter permits even higher spinning rates. Additional benefits of spherical rotors include easy sample exchange and improved microwave access for dynamic nuclear polarization (DNP)-NMR experiments,



yielding improved microwave $B_1$ field strength and homogeneity compared with current methods (Chen et al. (2019); Gao et al. (2019)).

      In order to make spherical rotors robust and accessible for magnetic resonance experiments and to design apparatuses capable of achieving very high MAS rates, we examined the spinning and stabilization mechanisms of these spherical rotors. Here we: (i) demonstrate how a stator for spinning spheres can be easily integrated into a commercial NMR probehead, and

(ii) examine the spinning behavior of a series of spherical rotors with various turbine groove geometries. We show that the spinning performance of spherical rotors can be improved by using a turbine groove geometry similar to the drive tips used in conventional cylindrical rotor MAS systems, which are themselves based on the Pelton impulse turbine (Wilhelm et al. (2015)). However, we also find across a wide array of turbine styles that spinning performance is remarkably indifferent to the surface design and that even a rotor without turbine grooves can achieve stable, on-axis spinning. We show that a spherical rotor attains

its stability from its inherent shape and mass distribution (i.e., its inertia tensor) and that turbine grooves are not essential for stable spinning.

## 2    Experimental Apparatus

The experimental apparatus is depicted in Figure 1. The stator employed for spinning spherical rotors was designed to adapt into a double-resonance APEX-style Chemagnetics probe (built decades ago for spinning 7 mm diameter cylindrical rotors)

and 3D printed in clear acrylonitrile-butadiene-styrene (ABS) using either a ProJet MJP2500 3D printer (3D Systems, Rock Hill, SC, US) or a Form3 3D printer (Formlabs, Somerville, MA, US). A double saddle coil made from 1.5 mm silver-coated copper magnet wire was wound by hand using a mandrel and wrapped in Teflon tape for insulation with the leads soldered into place in the existing RF circuit of the Chemagnetics probe. The two optical fibers of the tachometer system were introduced into holes at the bottom of the semi-transparent stator. The transceiver of the original tachometer system was replaced with a more

sensitive circuit. The transmitter in this circuit was an SFH756 light emitting diode fed with 42 mA of current. The detector comprised an SFH250 photodiode and a 4.7 MΩ transimpedance amplifier followed by a gain block providing a voltage gain of 42. The magic angle adjustment was achieved by coupling the stator to the existing angle adjust rod in the probe.

      NMR experiments were performed at 7.05 T using a Bruker Avance III spectrometer (Bruker Corp., Billerica, MA, US). [79]Br spectra were taken at a transmitter frequency of 75.46 MHz and a MAS rate of 3.5 kHz. The implementation of a double

saddle coil within the probe enabled the application of a 35 kHz $B_1$ field on [79]Br with 300 W incident RF power, a significant improvement over our previous implementation of 9.5 mm spherical rotors, which achieved a $B_1$ field of only 12.5 kHz using a split coil and 800 W incident RF power Chen et al. (2018).

      The 9.5 mm spherical rotors were machined from yttria-stabilized zirconia (O'Keefe Ceramics, Woodland Park, CO, US). Seven new spherical rotor designs were introduced (Figure 2), each with a 2.54 mm inner diameter cylindrical through hole:

notched (rotors A, C), Pelton-style (rotors B, F), circular (rotors C, G), dimpled (rotor D), and with no flutes (rotor H). For the notched, circular, and Pelton-style rotors, two variations were machined differing by 0.5 mm in depth. For spin testing, each rotor was filled with a rigid 3D printed cylindrical ABS blank terminated with 4.75 mm spherical radius contoured ends. For



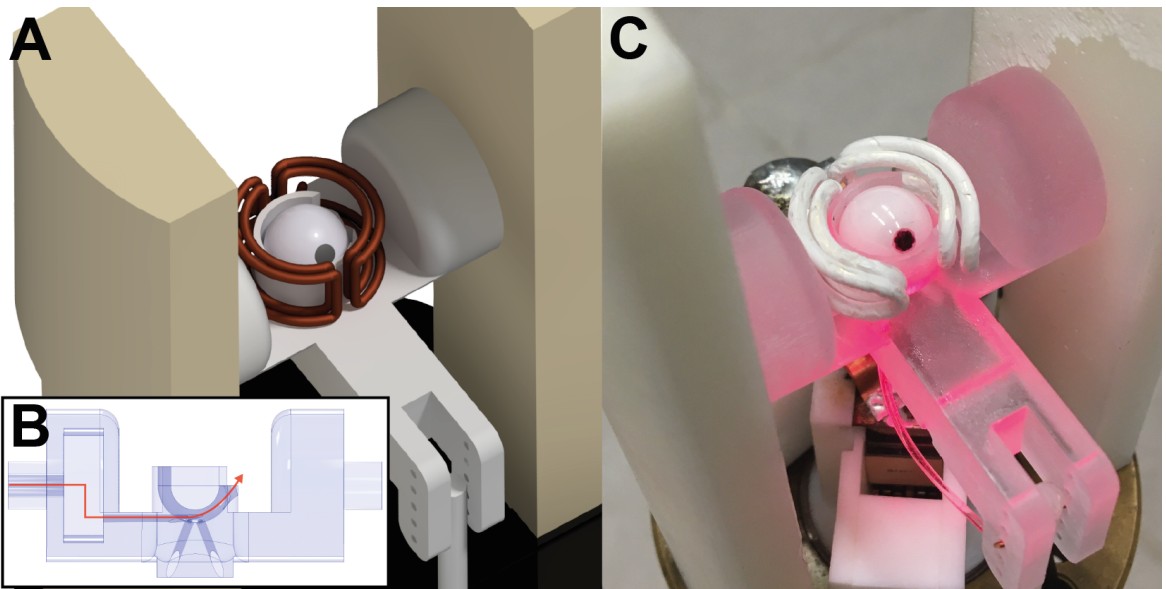

**Figure 1.** Experimental apparatus. A.) Probehead design schematic depicting the stator with a double saddle coil, spherical rotor, and angle adjust arm. B.) Cross-section view of the stator showing the drive gas inlet and angled through holes for the fiber optic tachometer. Drive gas is fed from the left side into the stator cup (red arrow). C.) Fully assembled probehead.

the NMR experiment, the rotor was packed with 97.1 mg KBr powder and sealed at each end with 3D printed ABS plugs. The same stator design was used for all spin testing experiments.

## 3 Results and Discussion

Spherical rotors spin within the hemispherical stator cup by the application of a gas stream along the rotor's equator from a converging nozzle tangent to the rotor surface (Chen et al. (2018)). The nozzle aperture is placed at the complement of the magic angle (35.26°) in order to tilt the spinning axis of the rotor to a value near the magic angle. The rotor turbine grooves are intended to provide a means to efficiently couple the gas stream to the rotor surface, converting the kinetic energy of the fluid flow into rotational kinetic energy.

The designs depicted in Figure 2 were chosen to explore this concept. Intuition might suggest that deeper grooves allow for greater coupling to the gas stream due to the increased surface area perpendicular to the direction of fluid flow. However, the deep-grooved rotors E, F, and G could not spin stably under any of the conditions tested. It is possible that the large spaces created by the removal of material to make these deep grooves allow chaotic and turbulent flows to develop within the stator cup. The shallow-grooved rotors A, B, C, and D, as well as rotor H, achieved stable spinning. Figure 3A shows the spin test data in air at gauge pressures ranging from 0 to 4.3 Bar for these five rotors. Across the five turbine geometries, the spinning rate increased non-linearly with respect to the applied pressure; pressure increases at higher pressures less effectively increased



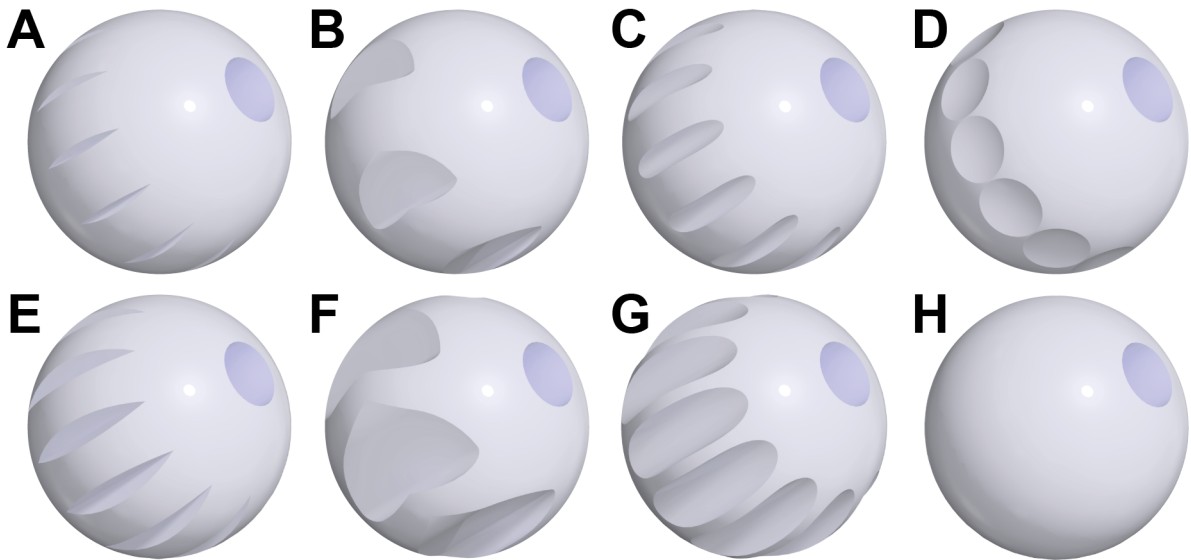

**Figure 2.** 9.5 mm spherical rotor turbine groove designs. Each rotor has a 2.54 mm diameter through hole. A.) Twelve 60º notched grooves, 0.35 mm depth (as described previously by Chen et al. (2018). B.) Six impeller-style grooves, 0.25 mm radius. C.) Twelve circular 0.25 mm radius grooves. D.) Twelve dimpled grooves, 4.75 mm radius dimples. E.) Twelve 60º notched grooves, 0.85 mm depth. F.) Six impeller-style grooves, 0.5 mm radius. G.) Twelve circular 0.5 mm radius grooves. H.) Smooth surface, no flutes machined.

the spinning rate than at lower pressures. The maximum spinning rates for the five rotors at the pressures tested ranged between 4 and 6 kHz in air, with Pelton-style rotor B having the highest maximum spinning rate of 5.7 kHz. Note that the maximum
tested pressure was not limited by the rotors, as their spinning rates would be predicted to increase with increasing pressure, but rather the maximum pressure was limited to ∼4 Bar due to safety concerns with regard to the connecting assemblies. Rotor B's maximum observed rate was a significant improvement over our previous maximum of 4.6 kHz for 9.5 mm rotors in air, which was achieved using rotor A at 4.1 Bar (Chen et al. (2018)). As rotor A also performed better in our current tests, with a maximum rate of 5.2 kHz at 4.1 Bar, we attribute some of the performance gains to the higher precision 3D printing of our latest
stators. However, the further increase to 5.7 kHz with rotor B is likely to be due to the Pelton-style grooves more efficiently coupling the fluid flow and the rotor surface with a sufficiently shallow groove profile to prevent undesirable complex fluid flows in the stator.

A significant finding was that rotor H, with its smooth surface and lack of grooves, spun stably and on-axis. While not delivering the highest frequency spinning of the tested rotors, its performance was comparable to many of the designs with
machined grooves. Figure 3B shows the $^{79}$Br spectrum of KBr at 3.5 kHz MAS using rotor H. The stator's pitch angle was adjusted until the rotor's spinning axis was inclined to the magic angle, taken as the maximal number of rotor echoes in the time domain data. The spectrum shows the spinning sideband manifold equally spaced by 3.5 kHz, corroborating the spinning rate observed by optical tachometry. The spinning was stable, with a standard deviation of ±1 Hz without the use of any spinning





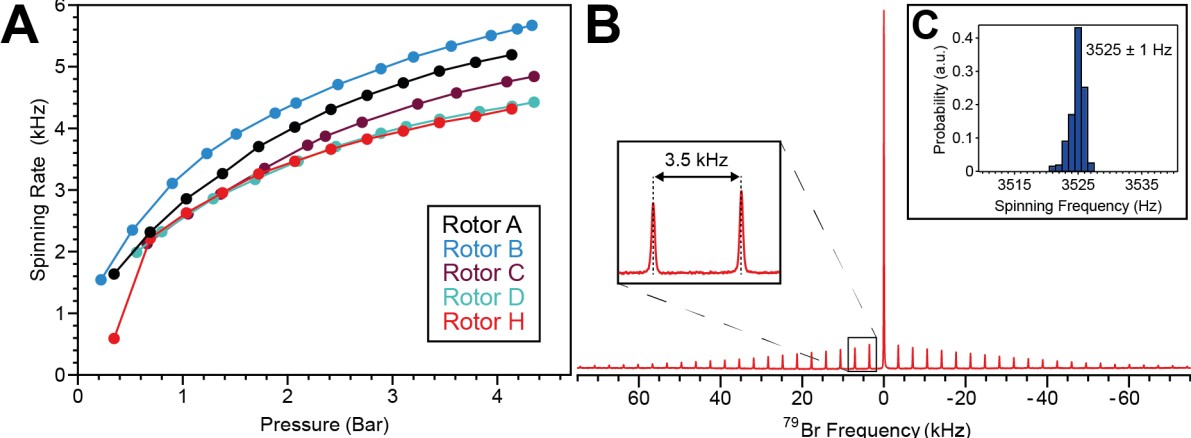

**Figure 3.** Spin test data. A.) Spinning rate as a function of applied air pressure for rotors A, B, C, D, and H. B.) $^{79}$Br spectrum of KBr packed into rotor H and spinning at 3.5 kHz at the magic angle; 512 scans. C.) Histogram of spinning frequencies without spinning regulation over the 10 minute KBr data acquisition period.

regulation mechanism (Figure 3C). The ability of rotor H to spin stably at reasonable MAS rates implies that while turbine

grooves can improve fluid flow and rotor coupling, a significant contribution to the overall spherical rotor spinning mechanism is simply from the torque created by drag induced by the driving gas stream moving across the rotor surface. Additionally, the fact that rotor H established a stable spinning axis about its own axis of symmetry shows that the grooves do not direct the rotor to spin about this axis, but rather the geometry of the rotor itself is responsible.

A spherical rotor with a cylindrical through hole is a solid known as a spherical ring. The inertia tensor for a spherical ring

of constant density $\rho$, outer radius $R$, and inner radius $r$, where $R \geq r$ and $z$ lies along the axis of cylindrical symmetry, is given by:

$$\boldsymbol{I_{sr}} = \frac{4}{15}\pi\rho(R^2 - r^2)^{3/2} \begin{pmatrix} \frac{4R^2+r^2}{2} & 0 & 0 \\ 0 & \frac{4R^2+r^2}{2} & 0 \\ 0 & 0 & 2R^2 + 3r^2 \end{pmatrix} \tag{1}$$

Considering most previous MAS experiments have been performed with cylindrical rotors, it is worth examining the inertia tensor of a cylindrical shell for comparison. For a cylindrical shell of constant density $\rho$, outer radius $R$, inner radius $r$, and

length of $2kR$, where $k$ is the aspect ratio and $R \geq r$, the inertia tensor is given by:

$$\boldsymbol{I_{cs}} = \pi\rho kR(R^2 - r^2) \begin{pmatrix} \frac{(2k+1)R^2+r^2}{2} & 0 & 0 \\ 0 & \frac{(2k+1)R^2+r^2}{2} & 0 \\ 0 & 0 & R^2 + r^2 \end{pmatrix} \tag{2}$$





Figure 4 shows the magnitudes of the moments of inertia for a spherical ring and a cylindrical shell ($k = 4$) as a function of the inner radius given by equations 1 and 2. For the spherical ring, $I_z$ is greater than or equal to the transverse moments $I_{x,y}$ for all values of $r$, while for the cylindrical shell, $I_z$ is less than or equal to the transverse moments $I_{x,y}$ for all values of 105 $r$. When $r = 0$, the moments of inertia for the spherical ring and cylindrical shell are equivalent to those of a solid sphere and solid cylinder, respectively. Note that while Figure 4B is representative of the high aspect ratio cylindrical rotors commonly used in MAS experiments, when $k$ is low such that the geometry is disk-like, $I_z$ will be greater than $I_{x,y}$ for all values of $r$.

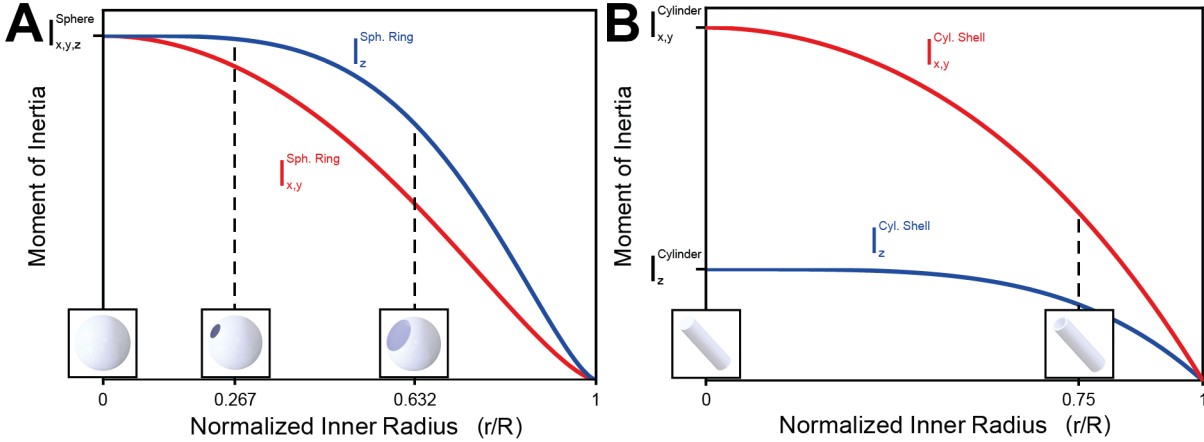

**Figure 4.** Moments of inertia for spherical and cylindrical rotors as a function of the normalized inner radius $r/R$. A.) $I_z$ (blue) and $I_{x,y}$ (red) for a spherical rotor as a function of the normalized inner radius. The rotors spun in this study correspond to $r/R = 0.267$. The difference between $I_z$ and $I_{x,y}$ is maximized at $r/R = 0.632$. B.) $I_z$ (blue) and $I_{x,y}$ (red) for a cylindrical rotor with aspect ratio $k = 4$ as a function of the normalized inner radius.

An object is capable of spinning stably about any axis without the need for stabilization as long as there are no avenues to dissipate rotational energy. However, when such dissipative elements are present, the object will end up rotating about the 110 axis that minimizes its rotational kinetic energy for a fixed angular momentum, which is the axis with the largest moment of inertia (Efroimsky (2001, 2002); Krechetnikov and Marsden (2007)). This phenomenon has been observed in objects such as comets and asteroids as well as in early spacecraft such as Explorer 1 (Efroimsky (2001); Krechetnikov and Marsden (2007); Bracewell and Garriott (1958)). Explorer 1 was a cylindrical satellite with a high aspect ratio that was meant to spin about its axis of symmetry (the lowest inertia axis) but ended up spinning end-over-end (the highest inertia axis) as a result of 115 energy being dissipated into the structure. Similarly, as a result of this phenomenon, asteroids are more often found tumbling end-over-end rather than spinning about an axis of cylindrical symmetry.

Like a cylindrical satellite, a high aspect ratio cylindrical MAS rotor requires active stabilization (i.e., a bearing) in order to spin stably about its axis of symmetry, and any instabilities in the spinning could magnify as the rotor attempts to transition into a spin about its highest inertia axis, potentially leading to a rotor crash. However, in the case of a spherical MAS rotor,



**MAGNETIC RESONANCE**
Discussions
the rotor is placed into the stator without a specific orientation and initially may spin about an arbitrary axis. As the axis of symmetry for a spherical ring is also its greatest inertia axis, a spherical rotor ultimately ends up stably spinning about its axis of symmetry due to rotational energy being dissipated by interactions with the gas stream and surrounding atmosphere. For this reason, a spherical rotor shows very stable on-axis spinning and resilience to crashing, as any spinning instabilities will be damped by the surrounding fluid and cause the rotor to return to a stable minimum about its axis of symmetry.

Equation 1 suggests that spherical rotors should spin most stably about $z$ when $r/R = 0.632$, where the difference between $I_z$ and $I_{x,y}$ is maximized. Critically, this means that rotors with a spherical ring geometry can be quite stable even with very large sample volumes. When considering a packed rotor, as long as the density of the caps and sample are lower than the density of the rotor material, $I_z$ will be greater than $I_{x,y}$ for all values of $r/R$ between 0 and 1, and stable on-axis spinning will occur.

**4  Conclusions**

While turbine grooves can help to increase MAS rates for spherical rotors, the inertia tensor is responsible for its spinning stability. As MAS rates continue to increase to values in excess of 100 kHz, the possibility of spinning instabilities leading to rotor crashes becomes a significant concern. Spherical rotors spinning at high rates will be able to self-correct to a stable state after a perturbation due to the large moment of inertia about their axis of symmetry. To achieve high MAS rates with spherical

rotors, new rotors with smaller outer diameters must be designed and fabricated. These rotors could use shallow, Pelton-style grooves to increase the maximum spinning rate by about 30% compared to a rotor with no machined grooves, as observed here. However, since turbine grooves are not necessary to achieve stable spinning, these rotors could be fabricated without the need for complex micro-machining techniques to produce turbine grooves.

*Author contributions.* T.O.P. performed the stator, rotor, and probe design, spin test and NMR data collection, the inertia tensor analysis

and the writing of the manuscript. A.D. and C.G. fabricated and installed the double-saddle coil and optical tachometer. P.C. assisted with the design process, spin testing, and data collection. L.E.P. assisted with 3D printing of the stators. N.H.A. assisted with the rotor design process. A.B.B. supervised the execution of experiments and guided the writing of the manuscript. All authors contributed to the editing of the manuscript.

*Competing interests.* A.B.B. is an author on provisional patents related to this work filed by Washington University in Saint Louis (62/703,278

filed on 25 July 2018 and 62/672,840 filed on 17 May 2018). The authors declare no other competing interests.

*Acknowledgements.* This research was supported by an NIH Director's New Innovator Award [grant number DP2GM119131] and start-up funding from ETH Zürich. We thank Roland Walker for valuable assistance and advice with 3D printing.





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
