# Peer review of "Highly Stable Magic Angle Spinning Spherical Rotors"

_Magnetic Resonance, 2020_

## Short Comment (SC1) · 7 Apr 2020

It would be interesting to know if more than 5 rotors were tested, and if the performance documented in Fig. 3 refers to their average performance or to unique cases.

The description of the curves of Fig. 3 as "pressure increases at higher pressures less effectively increased at spinning rate than at lower pressure" should be replaced by a sober reference to the figure itself.

While the discussion of the moments of inertia for empty spherical and cylindrical rotors is interesting, it is only at the very end that the authors admit that the sample and caps will affect these considerations.

The sentence "adjusted until the rotor's spinning axis was inclined to the magic angle,

taken as the maximal number of rotor echoes in the time domain data" leaves me dumbfounded. Surely it is the decay of the envelope of the rotational echoes that could be taken as a measure for the adjustment of the magic angle.

The claim that "the fact that rotor H established a stable spinning axis about its own axis of symmetry" is interesting, but this hardly "shows that the grooves do not direct the rotor to spin about this axis, but rather the geometry of the rotor itself is responsible". There is no evidence that the machining of the groves agrees with the geometry.

I have learned in a first-year physics course (at ETH!) that objects end up tumbling around the axis with the largest moment of inertia. I believe that this has been known since the XIXth, perhaps even since the XVIIIth century. It is unfortunate that the designers of "early spacecraft such as Explorer 1" were not aware of this phenomenon.

This seems to be a rather hastily written progress report, in my opinion not suitable of MR.

---

## Referee Comment (RC1) · Anonymous Referee #1 · 10 Apr 2020

General Comments:

The authors discuss eight different turbine groove designs for spherical magic angle spinning rotors. They find that deep turbine grooves do not allow stable spinning, and that some shallow groove designs allow modest increases in spinning speed compared to a groove-free surface. The stability of these spherical ring rotors is discussed in terms of the rotor's principle moments of inertia, and compared to the situation for more conventional cylindrical MAS rotors.

The paper reports progress on the optimization of the very novel magic angle spinning rotor system design that has come out of this laboratory in recent years, and gives a theoretical basis for why the stability of this design is so robust.

[Figure]

The results reported here represent a necessary step in the evolution and optimization of this new spinning system design. While the community of researchers who build their own magic angle spinning systems is rather small, and those who spin with spherical rotors is smaller still, this work represents what I hope will be one of many modest forward steps that will eventually make spherical rotors a compelling alternative to conventional designs.

The paper is logically organized and easy to follow.

Specific comments:

The title seems inappropriate for the work described. While I understand that the fact that grooveless rotors perform nearly as well as the best grooved design is one of the significant results, the title ignores most of the experiments described.

In considering moments of inertia, the authors consider empty rotors: spherical rings or cylindrical shells. But some conventional cylindrical rotor designs do not spin well empty - the sample matters. The addition of the sample is considered only cursorily at the end of the manuscript. Presumably if the sample density is much less than the density of the rotor itself things aren't changed much by the sample, but maybe something more could be said?

The discussion of the stability of rotation is somewhat unsatisfying. There is a commonly known theorem about rotation that for objects with three distinct moments of inertia, rotations about the axes having the largest and smallest moments are stable, while rotation about the intermediate axis is not (tennis rackets are a prototypical example). That theorem would suggest that cylinders rotating about their long axes should be stable, as long as both energy and angular momentum are conserved. The situation with both spherical rings and cylinders might be a little different because of the cylindrical symmetry, where there is no intermediate axis. I'd like to see a bit deeper discussion of the stability criteria. While this represents old physics, it would be nice to see a sound discussion in the context of magic angle spinning systems.

I wonder if the statement on line 108, that rotation about any axis is stable if there is no energy dissipation, is actually helpful in understanding stability issues?

Minor issues:

pg 2 line 46. What is meant by 4.7 M-ohm transimpedance amplifier? Does that mean a 4.7 M-ohm resistor in series with the photodiode?

line 51 reference should be parenthesized.

line 162, 175 and others: links to cited doi's appear twice in a number of the references.

---

## Referee Comment (RC2) · Anonymous Referee #2 · 14 Apr 2020

General Comments:

The authors examine and discuss factors important for fast, stable spinning of spherical rotors for magic angle spinning solid-state NMR experiments. They show that a stator for spherical rotors can be integrated into a commercial NMR probehead in a straightforward manner. They present measurements of spinning speeds and spinning stability achieved with eight different turbine groove designs, including one design with no grooves. The spinning rate achieved at a given pressure is shown to be sensitive to the surface grooves, with moderate spinning rate improvements observed for some groove geometries over others, and over a smooth rotor. Finally, they present a theoretical motivation for the spinning stability they observed in spherical rotors based on an analysis of the principal values of a spherical rotor's moment of inertia tensor.
The results presented here are broadly interesting to the solid-state NMR community. Spherical rotors represent an alternative to the conventionally used cylindrical rotors, with several potential benefits which the authors outline. While much development work is still necessary, this work reports progress which is likely to spur further investigation into spherical rotors, and into other alternative rotor designs.

Specific Comments:

It would be interesting to hear comments about how spherical rotor stability is influenced by the sample, beyond what was included at the end of the manuscript. NMR data acquired on a sample spun in rotor H were acquired with a 3.5 kHz spinning speed, which is below the maximum spinning speed reported for rotor H in Figure 3a. Was 3.5 kHz chosen for stability reasons?

Was any consideration given when designing deep turbine grooves to how these grooves might alter the moment of inertia tensor of the spherical rotors?

Technical corrections:

The formatting of several of the references is somewhat strange, with the doi appearing twice in multiple references.

---

## Editor Comment (EC1) · Robert Tycko (Editor) · 16 Apr 2020

This manuscript is potentially suitable for publication in Magnetic Resonance after the authors make revisions that fully address the comments of the two anonymous reviewers as well as the following points:

1. The "tennis racquet theorem" says that any object in free space will rotate stably about axes close to its smallest and largest principal axes of inertia, but not stably around the middle axis of inertia. This is also mentioned by one of the anonymous reviewers. The authors should explain how this relates to their statement that "an object is capable of spinning stably about ANY axis ... as long as there are no avenues to dissipate rotational energy." The authors' statement seems erroneous, except perhaps

because they are discussing situations where the object is not in free space. Their later statement that "a high aspect ratio cylindrical MAS rotor requires active stabilization...in order to spin stably about its axis of symmetry" also appears to contradict the tennis racquet theorem.

2. The examples of satellites and asteroids may not be relevant to an MAS rotor. I suspect the behavior of satellites and asteroids may be affected by INTERNAL dissipation (movement of internal material), which is not an issue for an MAS rotor. This may need clarification.

3. A potential problem with spherical rotors may be that the magic angle needs to be readjusted for each sample, in other words the final direction of the axis of rotation may depend on the mass distribution within the rotor or on imperfections in the rotor itself. Is this true? The authors should comment on this issue, one way or the other.

4. The description that "the nozzle aperture is placed at the complement of the magic angle in order to tilt the spinning axis of the rotor to a value near the magic angle" needs further clarification. A more detailed drawing of the stator in Figure 1 might help.

---

## Short Comment (SC2) · 20 Apr 2020

I have two concerns regarding this draft.

1. Figure 4A shows that as the inner radius (r) in a sphere increases, the moment of inertia along Iz and Ix(or Iy) become unequal, with inertia along Iz being larger than Ix. The authors have used this fact to support the spinning-stability of sphere without groves.

The moment of inertia of a sphere is proportional to the radius of sphere (R). Therefore, the absolute difference between the inertia in two directions (z and x) is also proportional to R.

In the case presented, the sphere is large 9.5mm (with the maximum speed of  ${\sim}4\text{-}5$

kHz), and therefore it has a preferred axis of rotation. But as one goes for smaller sphere to achieve faster spinning (which is a major goal here), the absolute difference between Iz and Ix will be smaller and smaller. And in such scenario, the spinning along any particular axis will not be stable.

2. The authors have compared two cases, a sphere vs. a cylinder in figure 4. Sphere (hollow one) has preference to spin along "z" and a cylinder along "x". Using this simple comparison, authors have shown why a sphere is better than cylinder for stability.

In reality, the sample cup should be viewed as a combination of coaxial: (i). Spherical ring, (ii) a hollow cylinder (in which sample will be filled), (iii) a solid cylinder (basically the sample filled in the cylinder) and (iv) curved cap.

Different components (i, ii, iii and iv) will have different inertia. These are known or at least easy to calculate. Since moments of inertia are additive, it is possible to do a a more realistic calculations, taking into consideration moments of inertia of all these components.

Minor concern: Authors have used same notations to represent the dimensions of the two objects. It is better to use distinguished symbols to , e.g., r\_s, R\_s, r\_c and R\_c.

---

## Author Comment (AC1) · 7 May 2020

Response to Geoffrey Bodenhausen

Comments are in black, responses are in blue

It would be interesting to know if more than 5 rotors were tested, and if the performance documented in Fig. 3 refers to their average performance or to unique cases.

We tested all 8 rotors, however rotors E, F and G did not spin stably. Figure 3 refers to unique cases, but the results are reproducible to within ~100 Hz for each rotor at each pressure value. Collecting statistics is useful for our development of these rotors and stators, but the results can vary based on which printed stator is used and the exact conditions of the experimental setup. The figure shown represents a unique test series performed using the same stator and experimental conditions for all rotors, as described in the experimental details.

The description of the curves of Fig. 3 as "pressure increases at higher pressures less effectively increased at spinning rate than at lower pressure" should be replaced by a sober reference to the figure itself.

We agree that this sentence does not benefit the discussion and have removed it.

While the discussion of the moments of inertia for empty spherical and cylindrical rotors is interesting, it is only at the very end that the authors admit that the sample and caps will affect these considerations.

To address this concern, as other reviewers have also noted interest in this issue, we have added as supplementary material an interactive Mathematica document which allows the reader to independently adjust the densities for the sample, caps, and rotor in order to see the effect on the moments of inertia as a function of normalized inner radius. We have added additional discussion on this topic to this document. The model we use is a simple approximation of how we pack sample into the rotors, but should give a sense for the effects of sample and cap density on the moments of inertia of the overall packed rotor.

The sentence "adjusted until the rotor's spinning axis was inclined to the magic angle, taken as the maximal number of rotor echoes in the time domain data" leaves me dumbfounded. Surely it is the decay of the envelope of the rotational echoes that could be taken as a measure for the adjustment of the magic angle.

In a practice, when setting the magic angle, the researcher usually looks to see if more echoes are visible beyond the noise level out to a certain time in the time domain data as the angle is adjusted. Perhaps we chose a bit too practical of a description. We have modified this sentence to read: "The stator's pitch angle was adjusted until the rotor's spinning axis was inclined to the magic angle, as observed by measuring the decay of rotor echoes out to 10 ms in the time domain data."

The claim that "the fact that rotor H established a stable spinning axis about its own axis of symmetry" is interesting, but this hardly "shows that the grooves do not direct the rotor to spin about this axis, but rather the geometry of the rotor itself is responsible". There is no evidence that the machining of the groves agrees with the geometry.

We have measured the rotors that were machined, and found them to be within the tolerance levels of our design specifications. If a rotor with no grooves spins stably about its own axis, it shows that the grooves are not required in order to spin stably. The reason

it spins stably about its own axis must be due to another factor, and that is the spherical-ring geometry of the rotor itself.

I have learned in a first-year physics course (at ETH!) that objects end up tumbling around the axis with the largest moment of inertia. I believe that this has been known since the XIXth, perhaps even since the XVIIIth century. It is unfortunate that the designers of "early spacecraft such as Explorer 1" were not aware of this phenomenon.

This is a very important issue, and there is a lot of confusion surrounding this topic (including for us as we have worked on writing and developing this manuscript!), as classical rotational dynamics is an issue which seems simple on first glance, but is actually quite complex and has received significant attention and development over the past century. Euler's equations have been known since the 18th century, but they do not say anything about objects ultimately ending up spinning about the axis with the greatest moment of inertia. They predict that a rigid, axially-symmetric object will spin stably about its axis of symmetry regardless of whether the moment of inertia along that axis is the greatest or smallest moment. What happened with the Explorer 1 was that the satellite was intended to spin about its lowest inertia axis, which should have been fine if the object was rigid. However, due to avenues for dissipating rotational energy internally being present (antennas on the craft) the satellite began a precession which eventually turned into an end over end spin. The result of Explorer 1 spurred the further development of rigid-body rotational dynamics to account for this phenomenon. If avenues to dissipate rotational energy are present, the "stable" configuration initially predicted by Euler's equations is no longer stable. In MAS, a cylindrical rotor exchanges energy with the surrounding gas and can enter into a precession if it is not actively kept from doing so. This is partly why oversized bearings in the stator result in unstable spinning.

---

## Author Comment (AC2) · 7 May 2020

Response to Anonymous Referee #1

Reviewer comments are in black, responses are in blue

General Comments:

The authors discuss eight different turbine groove designs for spherical magic angle spinning rotors. They find that deep turbine grooves do not allow stable spinning, and that some shallow groove designs allow modest increases in spinning speed compared to a groove-free surface. The stability of these spherical ring rotors is discussed in terms of the rotor's principle moments of inertia, and compared to the situation for more conventional cylindrical MAS rotors.

The paper reports progress on the optimization of the very novel magic angle spinning rotor system design that has come out of this laboratory in recent years, and gives a theoretical basis for why the stability of this design is so robust.

The results reported here represent a necessary step in the evolution and optimization of this new spinning system design. While the community of researchers who build their own magic angle spinning systems is rather small, and those who spin with spherical rotors is smaller still, this work represents what I hope will be one of many modest forward steps that will eventually make spherical rotors a compelling alternative to conventional designs.

The paper is logically organized and easy to follow.

Specific comments:

The title seems inappropriate for the work described. While I understand that the fact that groove less rotors perform nearly as well as the best grooved design is one of the significant results, the title ignores most of the experiments described.

While we initially wanted to highlight the most interesting result in our title, we agree that the title as written does have the potential to overshadow the other experiments and discussion. Unless there are restrictions on changing the title after the discussion period, we have proposed a title change to "Highly Stable Magic Angle Spinning Spherical Rotors," in order to de-emphasize the focus on the rotor lacking turbine grooves and instead focus more broadly on the discussion of stability.

In considering moments of inertia, the authors consider empty rotors: spherical rings or cylindrical shells. But some conventional cylindrical rotor designs do not spin well empty - the sample matters. The addition of the sample is considered only cursorily at the end of the manuscript. Presumably if the sample density is much less than the density of the rotor itself things aren't changed much by the sample, but maybe something more could be said?

To address this concern, we have added as supplementary material an interactive Mathematica document which allows the reader to independently adjust the densities for the sample, caps, and rotor in order to see the effect on the moments of inertia as a function of normalized inner radius. We have added additional discussion on this topic to this document.

The discussion of the stability of rotation is somewhat unsatisfying. There is a commonly known theorem about rotation that for objects with three distinct moments of inertia, rotations about the axes having the largest and smallest moments are stable, while

rotation about the intermediate axis is not (tennis rackets are a prototypical example). That theorem would suggest that cylinders rotating about their long axes should be stable, as long as both energy and angular momentum are conserved. The situation with both spherical rings and cylinders might be a little different because of the cylindrical symmetry, where there is no intermediate axis. I'd like to see a bit deeper discussion of the stability criteria. While this represents old physics, it would be nice to see a sound discussion in the context of magic angle spinning systems.

We have taken this comment to heart and have now added a discussion with respect to the conditions of stability associated with axially symmetric objects. Using Euler's equations, one can show that a rigid, axially-symmetric object spins stably about its axis of symmetry regardless of whether the moment of inertia about that axis is the greatest or smallest moment. However, due to energy dissipation phenomena, objects tend to prefer to rotate about the axis with the highest moment of inertia.

I wonder if the statement on line 108, that rotation about any axis is stable if there is no energy dissipation, is actually helpful in understanding stability issues?

We have removed this statement in favor of a more rigorous discussion of the rotational dynamics.

Minor issues:

pg 2 line 46. What is meant by 4.7 M-ohm transimpedance amplifier? Does that mean a 4.7 M-ohm resistor in series with the photodiode?

The resistor in this case is in the feedback loop of the amplifier. The size of the feedback resistor relates to the gain of the amplifier and also determines the noise of the amplifier. We have adjusted the text to say "a transimpedance amplifier with a 4.7 M-Ohm feedback resistor" for clarity.

line 51 reference should be parenthesized.

This has now been corrected.

line 162, 175 and others: links to cited doi's appear twice in a number of the references.

We have now corrected these citations.

---

## Author Comment (AC3) · 7 May 2020

Response to Anonymous Referee #2

Reviewer comments are in black, responses are in blue

General Comments:

The authors examine and discuss factors important for fast, stable spinning of spherical rotors for magic angle spinning solid-state NMR experiments. They show that a stator for spherical rotors can be integrated into a commercial NMR probe head in a straightforward manner. They present measurements of spinning speeds and spinning stability achieved with eight different turbine groove designs, including one design with no grooves. The spinning rate achieved at a given pressure is shown to be sensitive to the surface grooves, with moderate spinning rate improvements observed for some groove geometries over others, and over a smooth rotor. Finally, they present a theoretical motivation for the spinning stability they observed in spherical rotors based on an analysis of the principal values of a spherical rotor's moment of inertia tensor.

The results presented here are broadly interesting to the solid-state NMR community. Spherical rotors represent an alternative to the conventionally used cylindrical rotors, with several potential benefits which the authors outline. While much development work is still necessary, this work reports progress which is likely to spur further investigation into spherical rotors, and into other alternative rotor designs.

Specific Comments:

It would be interesting to hear comments about how spherical rotor stability is influenced by the sample, beyond what was included at the end of the manuscript.

To address this concern, we have added as supplementary material an interactive Mathematica document which allows the reader to independently adjust the densities for the sample, caps, and rotor in order to see the effect on the moments of inertia as a function of normalized inner radius. We have added additional discussion on this topic to this document.

NMR data acquired on a sample spun in rotor H were acquired with a 3.5 kHz spinning speed, which is below the maximum spinning speed reported for rotor H in Figure 3a. Was 3.5 kHz chosen for stability reasons?

As shown in Figure 3A, rotor H is capable of spinning stably at 4 kHz at 4 bar of air pressure. However, we exercised reasonable caution for the NMR experiment, as 3.5 kHz can be achieved with nearly half the air pressure required to reach 4 kHz, and the experimental results achieve the same goal.

Was any consideration given when designing deep turbine grooves to how these grooves might alter the moment of inertia tensor of the spherical rotors?

At the time of their design, we were not considering that removing material might have a significant effect on the inertia tensor of the rotor itself, and instead assumed that the grooves would be a negligible change. However, it is possible that removing a significant amount of material from the outer equatorial region could have had an impact on their stability by changing the inertia tensor. We still maintain however, that the primary effect causing rotors E, F, and G to not spin stably was due to the excess space allowing for complex and turbulent fluid flows in the stator cup. The motion of rotors E, F and G in the

stator is random and chaotic even at very low pressures/flow rates, while for the other rotors, they simply don't spin until a certain pressure threshold is reached and then beyond that, spin up smoothly. More research is needed to explore the fluid dynamics of the gas flow within the stator cup, as even for the simplest case (Rotor H) we expect the fluid flow to be very complex and require a complete 3D description. For now, the guidance on turbine groove development is to make them as shallow as possible, which should also minimize changes to the spherical ring inertia tensor.

Technical corrections:

The formatting of several of the references is somewhat strange, with the doi appearing twice in multiple references.

We have now corrected this issue.

---

## Author Comment (AC4) · 7 May 2020

Response to Robert Tycko (Editor)

Reviewer comments are in black, responses are in blue

This manuscript is potentially suitable for publication in Magnetic Resonance after the authors make revisions that fully address the comments of the two anonymous reviewers as well as the following points:

1. The "tennis racquet theorem" says that any object in free space will rotate stably about axes close to its smallest and largest principal axes of inertia, but not stably around the middle axis of inertia. This is also mentioned by one of the anonymous reviewers. The authors should explain how this relates to their statement that "an object is capable of spinning stably about ANY axis ... as long as there are no avenues to dissipate rotational energy." The authors' statement seems erroneous, except perhaps because they are discussing situations where the object is not in free space. Their later statement that "a high aspect ratio cylindrical MAS rotor requires active stabilization...in order to spin stably about its axis of symmetry" also appears to contradict the tennis racquet theorem.

This statement was in error. This was meant to say "an axially-symmetric object is capable of spinning stably about any axis," but it turns out that this statement is also erroneous. We have removed this statement and instead included in the manuscript a discussion with respect to the conditions of stability associated with axially symmetric objects. Using Euler's equations, one can show that a rigid, axially-symmetric object spins stably about its axis of symmetry regardless of whether the moment of inertia about that axis is the greatest or smallest moment. However, due to energy dissipation phenomena, objects tend to prefer to rotate about the axis with the highest moment of inertia.

2. The examples of satellites and asteroids may not be relevant to an MAS rotor. I suspect the behavior of satellites and asteroids may be affected by INTERNAL dissipation (movement of internal material), which is not an issue for an MAS rotor. This may need clarification.

Avenues to dissipate energy in the vacuum of space must necessarily concern internal dissipation, as there is no surrounding medium to which the rotational energy can be dissipated. However, in the MAS stator, the surrounding gas dissipates rotational energy. The operating principle of pneumatic MAS depends on energy transfer between the gas and the rotor. We have clarified this point in our discussion.

3. A potential problem with spherical rotors may be that the magic angle needs to be readjusted for each sample, in other words the final direction of the axis of rotation may depend on the mass distribution within the rotor or on imperfections in the rotor itself. Is this true? The authors should comment on this issue, one way or the other.

Once adjusted to the proper angle for one rotor, a second rotor will spin at an angle very close to the magic angle, but not exactly. This is definitely a current challenge with the method, and something we are currently working on addressing. For now, it is recommended that all samples include some KBr to readjust the angle as needed, but we plan to solve this issue in an upcoming manuscript.

4. The description that "the nozzle aperture is placed at the complement of the magic angle in order to tilt the spinning axis of the rotor to a value near the magic angle" needs further clarification. A more detailed drawing of the stator in Figure 1 might help.

Figure 1 has been updated to show a cross section of the stator from another angle. We hope this addresses the concern.

---

## Author Comment (AC5) · 7 May 2020

Response to Asif Equbal

Comments are in black, responses are in blue

I have two concerns regarding this draft.

1. Figure 4A shows that as the inner radius (r) in a sphere increases, the moment of inertia along Iz and Ix(or Iy) become unequal, with inertia along Iz being larger than Ix. The authors have used this fact to support the spinning-stability of sphere without groves. The moment of inertia of a sphere is proportional to the radius of sphere (R). Therefore, the absolute difference between the inertia in two directions (z and x) is also proportional to R. In the case presented, the sphere is large 9.5mm (with the maximum speed of~4-5 kHz), and therefore it has a preferred axis of rotation. But as one goes for smaller sphere to achieve faster spinning (which is a major goal here), the absolute difference between Iz and Ix will be smaller and smaller. And in such scenario, the spinning along any particular axis will not be stable.

All calculations were performed without using a fixed value for the radius. The results depicted in Figure 4 are valid for any given radius- note that the x-axis is "normalized inner radius," which takes into account the fact that these results scale for any outer radius value. It is true that the absolute values of the moments of inertia will decrease with smaller rotors, but the ratio of values between Iz and Ix will always be the same. The size of the rotor should not affect these stability considerations. For experimental proof of this, we have spun 4mm spherical rotors stably at 28 kHz, and 2mm rotors at around 60 kHz (unpublished).

2. The authors have compared two cases, a sphere vs. a cylinder in figure 4. Sphere(hollow one) has preference to spin along "z" and a cylinder along "x". Using this simple comparison, authors have shown why a sphere is better than cylinder for stability. In reality, the sample cup should be viewed as a combination of coaxial: (i). Spherical ring, (ii) a hollow cylinder (in which sample will be filled), (iii) a solid cylinder (basically the sample filled in the cylinder) and (iv) curved cap. Different components (i, ii, iii and iv) will have different inertia. These are known or at least easy to calculate. Since moments of inertia are additive, it is possible to do a more realistic calculation, taking into consideration moments of inertia of all these components.

To address this concern, as other reviewers have also noted interest in this issue, we have added as supplementary material an interactive Mathematica document which allows the reader to independently adjust the densities for the sample, caps, and rotor in order to see the effect on the moments of inertia as a function of normalized inner radius. We have added additional discussion on this topic to this document. The model we use is a simple approximation of how we pack sample into the rotors, but should give a sense for the effects of sample and cap density on the moments of inertia of the overall packed rotor.

Minor concern: Authors have used same notations to represent the dimensions of the two objects. It is better to use distinguished symbols to , e.g., r_s, R_s, r_c and R_c.

For the results to be compared between the sphere and cylinder for a given outer radius R and inner radius r, we have chosen not to demarcate these with distinguished symbols.